# Combining Model-Agnostic Meta-Learning and Transfer Learning for Regression

**DOI:** 10.3390/s23020583

**Published:** 2023-01-04

**Authors:** Wahyu Fadli Satrya, Ji-Hoon Yun

**Affiliations:** 1Computer Science Department, BINUS Online Learning, Bina Nusantara University, West Jakarta, Jakarta 11480, Indonesia; 2Department of Electrical and Information Engineering, Seoul National University of Science and Technology, Seoul 01811, Republic of Korea

**Keywords:** meta-learning, regression, model-agnostic meta-learning, MAML, few-shot learning, transfer learning, model adaptation, ensemble

## Abstract

For cases in which a machine learning model needs to be adapted to a new task, various approaches have been developed, including model-agnostic meta-learning (MAML) and transfer learning. In this paper, we investigate how the differences in the data distributions between the old tasks and the new target task impact performance in regression problems. By performing experiments, we discover that these differences greatly affect the relative performance of different adaptation methods. Based on this observation, we develop ensemble schemes combining multiple adaptation methods that can handle a wide range of data distribution differences between the old and new tasks, thus offering more stable performance for a wide range of tasks. For evaluation, we consider three regression problems of sinusoidal fitting, virtual reality motion prediction, and temperature forecasting. The evaluation results demonstrate that the proposed ensemble schemes achieve the best performance among the considered methods in most cases.

## 1. Introduction

Meta-learning [1,2]—which refers to learning the foundational structures of a problem, i.e., learning to learn—has recently received much attention in the field of machine learning as a means of few-shot learning [3,4], i.e., learning a new task from only a few examples. When learning is initiated from scratch, ensuring that a neural network can learn a target problem sufficiently well generally requires a large number of data samples and a high computational cost. To realize few-shot learning, a meta-learning algorithm should be able to extract a meaningful statistical structure from past tasks, thus allowing the model to quickly adapt to a new task based on only a few data samples.

Model-agnostic meta-learning (MAML) [5] is a notable gradient-based framework of meta-learning. The virtues of MAML are its simplicity and the fact that it is applicable to a wide range of models as long as the gradient can be estimated because it relies solely on gradient-based optimization, with few configuration parameters and no need for model modification. Because of these advantages, MAML has drawn considerable attention, and several recent studies have sought to further improve its performance and stability [6].

The most straightforward strategies for adapting a model to a new target task (or dataset) are training from scratch (TFS) and training on everything (TOE) [5]. TFS involves reinitializing the model parameters and then retraining the model using only the new dataset for the target task. TOE involves reinitializing the model parameters and then retraining the model using all available data, including both old and new datasets. TOE can extract all available information from both the previous and new tasks. However, it may lead to negative transfer if the new task is greatly different from the previous tasks. On the other hand, TFS can avoid negative transfer, but it requires a large amount of data from the new task. Another approach to adaptation is transfer learning, which we call joint training (JT) in this paper. JT refers to further training a model using the new dataset for the target task without first reinitializing the model parameters. That is, JT merely fine-tunes the model parameters for the target task and thus gains the benefits of both TOE and TFS [7]. In [6], MAML was numerically compared with TFS, TOE and JT; however, the impact of the differences in the data distributions between the old and new tasks has not yet been explored.

In this work, we investigate how the data distribution differences between the old tasks and the new target task impact performance in regression problems. Through experiments, we discover that such differences greatly affect the relative performance of different adaptation methods. Our findings also reveal that as the distributional gap increases, MAML performs better than JT, whereas when the gap is small, JT performs better than MAML. Based on this observation, we develop an ensemble scheme combining multiple adaptation methods, especially MAML and JT, that can handle a wide range of data distribution differences between the old and new tasks, thus making it more stable for a wide range of tasks. For evaluation, we consider three regression problems of sinusoidal fitting, virtual reality (VR) motion prediction, and temperature forecasting. The evaluation results demonstrate that the proposed ensemble scheme achieves the best performance among the considered methods in most cases, i.e., 100% of cases for the sinusoidal regression problem, 75% of cases for the VR motion prediction problem, and 100% of cases for the temperature forecasting problem.

The rest of this paper is organized as follows. Related studies are reviewed and discussed in Section 2. In Section 3, we describe the MAML framework. Section 4 presents important observations about the impact of data distribution differences based on the results for the sinusoidal regression problem. The proposed ensemble scheme is described in Section 5. The setup and performance results of comparative experiments are provided in Section 6. Finally, Section 7 concludes the paper.

## 2. Related Work

Meta-learning. Meta-learning refers to learning a problem structure from a given set of tasks, called meta-tasks. After a model has learned the meta-tasks, it may learn a new task faster [8]. Several techniques for implementing meta-learning have been reported. A memory-augmented neural network [9] performs meta-learning by using a neural Turing machine to achieve faster encoding, retrieve new information, and solve the problem of conventional models in that their parameters need to be relearned. A convolutional Siamese neural network [10] combines a Siamese neural network and a convolutional architecture to implement meta-learning. It compares two sets of inputs, namely, test images and support images, based on the features of the images. The support image set contains training images along with their classes, and the network seeks to verify whether there is any similarity between the test and support images. Long short-term memory (LSTM)-based meta-learning was introduced in [11]. In this approach, an LSTM unit is used as a meta-learner to obtain an optimization, which is then used to train a new learner.

MAML is a meta-learning method that is applicable to various problems, such as regression, classification, and reinforcement learning [5]. The use of the first-order derivative has been recommended to enable faster training at the cost of sacrificing prediction performance. A first-order version of MAML was also proposed in [12]. It ignores second-order derivatives and still achieves good performance by means of a multistep inner loop. The gradient in the outer loop is computed with the last updated parameter. To determine an effective step size increase, all gradient steps in the inner loop are summed and averaged. MAML can also be used for online learning [7] by extending the MAML algorithm to the method known as ‘follow the meta-leader’ (FTML).

There have been various attempts to enhance MAML in terms of stability, computational cost, and applicability. Multiple issues leading to the performance degradation of MAML were identified in [6]. To overcome these issues, the authors proposed the use of cosine annealing, derivative-order annealing, batch normalization, per-layer per-step learnable learning rates, and multistep loss optimization. In the multistep loss approach [6], gradient degradation in MAML is addressed by computing the loss on the target set in every inner loop and thus directing the attention of the MAML model to the main goal. Cosine annealing [13] has been proven to help achieve state-of-the-art model performance. Derivative-order annealing [6] helps reduce the meta-training workload of MAML without affecting the final performance by utilizing first-order optimization in the first few meta-training steps. In the meta-curvature approach [14], additional trainable parameters are introduced to capture a common representation of the training tasks in the form of a curvature. Per-layer per-step learnable learning rates [6] address the decrease in generalization ability and convergence speed caused by a fixed learning rate for all parameters by allowing the MAML model to learn its own inner-loop learning rate schedule in each layer. A multimodal MAML framework was proposed in [15] to identify the modes of tasks sampled from a multimodal task distribution and quickly adapt gradient updates by using its meta-learned prior parameters in accordance with the identified mode. The authors of [16] reformulated the objective of MAML to minimize the maximum loss over the observed meta-training tasks to consider the worst-case task performance. In [17], the authors studied the generalization error properties of MAML for recurring and unseen tasks in supervised learning problems. To address the heterogeneity of the underlying data distributions for various users, the authors of [18] proposed a personalized variant of federated learning to find an initial shared model that can be easily adapted to the local dataset of either a current or new user by performing one or a few steps of gradient descent with respect to the user’s own data via MAML.

Transfer learning. Due to its broad application prospects, transfer learning has become a popular and promising area in machine learning, and several comprehensive surveys on transfer learning have been presented [19,20]. In conventional transfer learning, an entire new model is required for every task. The authors of [21] proposed performing transfer learning with adapter modules that yield a compact and extensible model by adding only a few trainable parameters per task and to allow new tasks to be added without revisiting previous ones. SpotTune [22] finds the optimal fine-tuning strategy per instance for the target data. Given an image from the target task, a policy network is used to make the routing decision, i.e., the decision of whether to pass the image through the fine-tuned layers or the pretrained layers. To fully utilize the categorical information of a pretrained model, the authors of [23] proposed a two-step transfer learning framework in which the relationship between the source categories and target categories is learned first and then the target labels are translated based on the learned category relationship and used to collaboratively supervise the fine-tuning process. In [24], the authors proposed a scheme for fast tuning in which not only are multiple tasks learned, but scaling and shifting functions for the neural network (NN) weights for each task are also learned.

Ensemble learning. Ensemble learning is a method of using multiple approaches by combining their results and then training the corresponding ensemble to achieve a better combined prediction performance than any of the individual approaches alone [25]. Several studies related to ensemble implementation have been reported. Ref. [26] used ensemble learning to make predictions for remote sensing classification. Ref. [27] implemented ensemble learning to predict building electricity usage. Ensemble learning has been shown to improve the prediction error performance of classification and regression trees (CART).

Several studies have used various techniques for ensemble learning. Ref. [28] used the random forest (bagging), extra trees (bagging), and gradient-boosted regression tree (boosting) algorithms as ensemble techniques to address data fusion in remote sensing. Among these techniques, different algorithms may have an advantage over the others depending on the dataset conditions. On the other hand, [29] used the stacking technique, which produces a final prediction by combining predictions from three base learning methods. The resulting ensemble system then produces a lower error than any of the individual weak base learning methods on the same dataset. Ref. [30] presented a combination of simple ensemble learners called multistrategy ensemble learning. This strategy shows better improvement than the individual simple ensemble learners because it achieves increased diversity without being affected by the loss of each simple ensemble learner.

## 3. MAML

MAML is composed of the following two training stages: meta-training and adaptation. The aim of the first stage, meta-training, is to learn the general structure of a given set of *M* tasks called meta-tasks and initialize a model with the generalized parameters that have been obtained. Each meta-task is associated with a dataset extracted from the corresponding task.

Let f(θ) be the model to be trained, with parameters θ. In the meta-training process, θ is evaluated for all meta-tasks and updated in the inner loop. For each meta-task $T_{i}$, MAML computes the adapted parameters $\theta_{i}^{{}^{\prime }}$ using the following gradient step:(1)$$\begin{array}{c} \theta_{i}^{{}^{\prime }}=\theta -\alpha \nabla_{\theta }L_{T_{i}}\left(f\left(\theta \right)\right) \end{array}$$
where $L_{T_{i}}$ is the loss function of $T_{i}$ and α is the learning rate in the inner loop. That is, $\theta_{i}^{{}^{\prime }}$ represents the parameters learned for $T_{i}$ using the gradient step of Equation (Equation 1). Once $\theta_{i}^{{}^{\prime }}$ has been obtained for each meta-task, θ is updated via the following process, which is called the outer loop:(2)$$\begin{array}{c} \theta \leftarrow \theta -\beta \nabla_{\theta }\sum_{i=0}^{M}L_{T_{i}}\left(f(\theta_{i}^{{}^{\prime }}\right)) \end{array}$$
where β is the learning rate in the outer loop. That is, θ is updated and the sum of the loss functions of the meta-tasks is reduced.

After the meta-training stage is completed, the process of adaptation to a new target task is performed. In the adaptation stage, the meta-parameters θ obtained from Equation (Equation 2) are used to perform further gradient steps, as expressed in Equation (Equation 1), for the dataset corresponding to the target task. Since the meta-parameters obtained from the meta-training stage are likely to provide a general structure for possible target tasks, only a small number of training samples for the actual target task and a few gradient steps should be necessary in the adaptation process to obtain the new adapted parameters.

## 4. Observation of the Impact of Differences in the Data Distributions between the Meta- and Target Tasks

We consider a simple regression problem in which each task involves regressing from the input to an output sine wave, where the amplitude and phase of the sinusoid vary between tasks, as illustrated in Figure 1 (regression to a specific curve corresponds to one task). Each task considered in the meta-training and adaptation stages has a sinusoidal input and output. To configure the distributional gap between the meta- and target tasks, we consider multiple experimental settings in which the data distributions differ in various ways, as shown in Table 1. Each setting is indexed by M$\left(m_{a},m_{p}\right)$-T$\left(t_{a},t_{p}\right)$, where M$\left(m_{a},m_{p}\right)$ indicates an amplitude distribution of $\left[0,2m_{a}\right]$ and a phase distribution of $\left[0,2m_{p}\pi \right]$ for the meta-tasks (i.e., $m_{a}$ and $m_{p}\pi$ are the mean amplitude and mean phase of the meta-tasks, respectively) and T$\left(t_{a},t_{p}\right)$ indicates the amplitude and phase configuration of $t_{a}$ and $t_{p}\pi$, respectively, for the target task. Accordingly, two types of distributional gaps between the meta-tasks and the target task can be defined:Phase distribution gap: Defined as $|t_{p}-m_{p}|\pi$. If $t_{p}<0$ or $t_{p}>2m_{p}$, the phase of the target task lies outside of the phase distribution of the meta-tasks.Amplitude distribution gap: Defined as $|t_{a}-m_{a}|$. If $t_{a}<0$ or $t_{a}>2m_{a}$, the amplitude of the target task lies outside of the amplitude distribution of the meta-tasks.

In the M(2.5, 0.25)-T(1, 0.75) setting, the phase of the target task lies outside of the phase distribution of the meta-tasks, corresponding to a higher phase distribution gap between the meta- and target tasks than in the other M(2.5, 0.25) settings. Similarly, M(2.5, 0.25)-T(1, 1) has the largest phase distribution gap among all M(2.5, 0.125) settings. A T(5, $t_{p}$) setting always has a larger amplitude distribution gap than a T(1, $t_{p}$) setting.

For MAML implementation, we used a second-order gradient with 1000 meta-training epochs and ten adaptation steps. The loss was defined as the mean-squared error (MSE) of prediction. The regression model consisted of two hidden layers, each with 40 neurons. In the optimization process, we used the Adam optimizer with learning rates of 0.01 for meta-training and 0.001 for adaptation. The frequency of all sinusoidal waves was set to 1/2π. (We tested different cases of the sinusoidal wave frequency but obtained trends similar to those for the considered frequency; therefore, the results for other frequency cases are omitted).

Figure 2 shows the results for the mean absolute error (MAE) gain of JT over MAML for an increasing phase distribution gap. A negative gain indicates that JT achieves a lower MAE than MAML; otherwise, MAML achieves a lower MAE than JT. For $m_{p}=0.25\pi$ (Figure 2a), the curve for a target amplitude of five monotonically increases; the curve for a target amplitude of one slightly decreases when the gap increases to 0.25 but ultimately increases when the gap further increases to 0.5. That is, with an increasing phase distribution gap, the MAE gain of JT over MAML also tends to increase. A similar trend is also observed for $m_{p}=0.125\pi$ (Figure 2b), although in this case, the gain always monotonically increases with an increasing phase distribution gap. The MAE gain of JT over MAML for an increasing amplitude distribution gap is shown in Figure 3. For the cases of both $m_{p}=0.25\pi$ and $m_{p}=0.125\pi$, it is clearly seen that the gain increases with an increasing amplitude distribution gap.

The above observations can be summarized as follows. When the target task is closer to the given meta-tasks (i.e., the distributional gap is smaller), the performance advantage of MAML over JT is smaller. However, if the target task is far from the given meta-tasks (i.e., the distributional gap is large), the performance advantage of MAML over JT is greater. If the distributional gap is sufficiently small, JT tends to outperform MAML, while if the gap is sufficiently large, MAML almost always outperforms JT.

## 5. Combining MAML and Transfer Learning

The observations in the previous section show that the performance of MAML depends on the gap between the data distributions of the meta- and target tasks. Similarly, other adaptation methods may also be affected by this distributional gap, but differently to MAML. Therefore, we propose combining MAML with other adaptation methods, an approach that we refer to as an ensemble scheme, to achieve better adaptation for a wider range of distributional gaps. The other adaptation methods included in the proposed ensemble scheme are illustrated in Figure 4 and are described below:Joint training (JT): As a pretraining step, the model is trained on all meta-tasks together as one large dataset. Then, the model is fine-tuned by using the dataset of the target task for adaptation.Training from scratch (TFS): The model parameters are randomly initialized. Then, the model is trained using the dataset of the target task. No pretraining using meta-task datasets is performed.Training on everything (TOE): The model is trained using all available data (datasets from both the meta- and target tasks). The pretraining and adaptation processes are not separate.

Ensemble learning uses the results from multiple models and combines them to improve the overall prediction performance. There are many types of ensemble learning. In the proposed ensemble scheme, the random forest (RF) method and gradient boosting (GB) [31] are used to implement ensemble learning. Any individual model, especially MAML or JT, may not perform well on the entire dataset but may work better than the others for certain parts of the dataset, as observed in the previous section. Thus, each model contributes to boosting the prediction performance in ensemble learning.

Figure 5 illustrates the architecture of the proposed ensemble scheme. The pseudocode of the scheme is given in Algorithm 1. The scheme consists of models at two levels. The first-level models are trained on datasets corresponding to the meta- and target tasks. The output results of the first-level MAML, JT, TOE, and TFS models then become the features of the dataset used by the second-level model. Before this dataset is processed, it is split into a training set and a test set. Before being processed by the ensemble algorithm, the training and test sets need to be converted into a data format that is compatible with the eXtreme Gradient Boosting (XGBoost) software library. For both RF and GB, we define a cross-validation function and optimize it using the Bayesian optimization (BO) package [32] to obtain the best set of hyperparameters, which is denoted by params. BO constructs a posterior distribution of functions (a Gaussian process) that best describes the loss function for the hyperparameters. As the number of observations grows, the Gaussian process is iteratively fitted to the known samples (previously explored hyperparameter values) and improved, and BO becomes more certain of which regions in the hyperparameter space are worth exploring. A new model is then trained using the best parameters and the training set. This model is called the second-level model. Finally, the second-level model is evaluated on the test set. This model can be represented as a tree and is updated every time a new iteration is completed. The tree model provides a correction for the error generated under the previous RF/GB parameters.
**Algorithm 1:** Ensemble scheme algorithm.1:$D_{train}$: Training dataset2:$D_{test}$: Test dataset3:params: Parameter set of the booster4:param_iter: Number of iterations for parameters5:init_point: Number of random exploration steps6:n_iter: Number of Bayesian optimization steps7:*N*: Number of RF/GB rounds8:**while **param_iter≤init_point+n_iter** do**9: params← Bayesian optimization results for a cross-validation function10:**end while**11:**while **n≤N** do**12: θ← ensemble.train(params, $D_{train}$)13:**end while**14:Evaluation of θ on $D_{test}$

## 6. Performance Evaluation

We evaluate the performance of the ensemble schemes (RF and GB) in comparison with MAML and the other adaptation methods using the following three regression problems: (1) sinusoidal regression (as described in Section 4), (2) VR motion prediction, and (3) temperature forecasting.

### 6.1. Sinusoidal Regression

We consider the same experimental setup described in Section 4. The methods are evaluated in terms of the MAE. Fifty meta-tasks and one target task are generated. The amplitude and phase distributions are configured as in Table 1. The numbers of meta-training and adaptation samples are both fixed at 600. For the pretraining stage in JT and the meta-training stage in MAML, 1000 iterations are specified. The number of adaptation steps is set to ten. The Adam optimizer is used with learning rates of 0.01 for pretraining/meta-training and 0.001 for adaptation. For the ensemble schemes, we combine the prediction results from the first-level models and divide them into the first 50% as the training set and the rest as the test set. For the Bayesian process, init_point is set to three steps, and n_iter is set to ten steps. For GB training, we use XGBoost [31] with ten boosting rounds performed to form a booster tree using GB.

Figure 6 shows the prediction performance under each experimental setting of the meta- and target tasks. For the settings with a target-task amplitude of one, the two ensemble schemes always achieve the lowest MAEs. JT achieves a lower MAE than MAML for the M(2.5, 0.25) and M(2.5, 0.125)-T(1, 0.125) settings but a higher MAE for M(2.5, 0.125)-T(1, 0.75) and an even higher MAE for M(2.5, 0.125)-T(1, 1). These results imply that using JT may result in a lower MAE if the distributional gap between the meta- and target tasks is small; using MAML is a better choice otherwise. This supports the benefit of the ensemble schemes, especially for MAML and JT. The results show that the data distributions do not greatly affect the ensemble schemes’ performance. TFS and TOE always have significantly higher MAEs than MAML. The pretraining stage of JT may lead to negative transfer from meta-training to adaptation if the gap between the data distributions is large. Thus, for the M(2.5, 0.125)-T(1, 0.75) and M(2.5, 0.125)-T(1, 1) settings, JT yields a worse result than TFS. For the settings with a target-task amplitude of five, the MAE is increased overall, but similar performance trends are still observed. The ensemble schemes still achieve significantly lower MAEs than the other methods.

### 6.2. VR Motion Prediction

It is known that the user-perceived quality of VR services strongly depends on the latency of the VR system. If the latency exceeds a certain threshold, the user will experience motion sickness or black borders that interfere with immersion. The latency problem is more severe in edge- or cloud-assisted VR services [33]. To combat the latency problem of a VR system, predictions of the future motion (head pose) of the user can be exploited [34].

We assume that we wish to predict the user’s head orientation at t+T from the information available by *t*. We denote the value of the variable *x* at time *t* by x[t] and use $x[t_{1},t_{2},\cdots ]$ to denote a time series of *x* at times $t_{1},t_{2},\cdots$. The information available at *t* can be represented as a window of sensor data samples reflecting the head orientation $p[t_{1},t_{2},\cdots ,t_{W}]$, the angular velocity $\dot{p}\left[t_{1},t_{2},\cdots ,t_{W}\right]$ obtained from a gyroscope, and the acceleration $\ddot{p}\left[t_{1},t_{2},\cdots ,t_{W}\right]$ obtained from an accelerometer, where *W* is the window size (number of data samples to be input), $t_{k}=t-\left(k-1\right)\tau$ with k=1,2,⋯,W, and τ is the time interval between consecutive data samples. Accordingly, the predicted head orientation, expressed as $\hat{p}\left[t+T\right]=\langle \hat{\theta }\left[t+T\right],\hat{\psi }\left[t+T\right],\hat{\phi }\left[t+T\right]\rangle$ (where $\hat{\theta }$, $\hat{\psi }$, and $\hat{\phi }$ are the pitch, roll, and yaw angles of the head orientation, respectively), can be defined as a function of the abovementioned sensor data. Thus, we formulate the VR motion prediction problem as follows:(3)$$\begin{array}{c} \hat{p}\left[t+T\right]=f_{\theta ,T}(p\left[t_{1},t_{2},\cdots ,t_{W}\right];\\ \;\dot{p}\left[t_{1},t_{2},\cdots ,t_{W}\right];\\ \;\ddot{p}\left[t_{1},t_{2},\cdots ,t_{W}\right]) \end{array}$$
where θ represents the model parameters of the function *f*. Then, the prediction error is defined as follows:(4)$$e\left[t+T\right]=\hat{p}\left[t+T\right]-p\left[t+T\right]$$

For the prediction of *K* samples, we calculate the MAE of prediction as follows:(5)$$\bar{e}=\frac{1}{K}\sum_{k=1}^{K}\left|e\left[k\tau \right]\right|$$

The function (model) *f* and its parameter set θ need to be found so as to minimize the MAE $\bar{e}$.

For the experimental evaluation of VR motion prediction, we collected three motion datasets using a VR headset for each of the following scenarios:Pitch motion in the upward direction;Pitch motion in the downward direction;Yaw motion in the rightward direction;Yaw motion in the leftward direction;Roll motion in the clockwise and counterclockwise directions;Playing VR Content 1;Playing VR Content 2;Playing VR Content 3.

The three VR contents are (1) moving around on the ground, (2) 360-degree cinematic video, and (3) an interactive game (fruit picking). Each dataset contains 300 data samples and is considered as a meta-task or a set of pretraining data. For the target tasks, seven users played VR Contents 1, 2, and 3.

We use a one-dimensional convolutional neural network (CNN) for prediction. For meta-training or pretraining in MAML and JT, the number of iterations is set to 200. The number of adaptation steps is set to ten. The Adam optimizer is used for both the pretraining and adaptation processes. In MAML, we set the learning rate to 0.002 for the inner loop and 0.01 for the outer loop, and the learning rate for adaptation is 0.005. In the ensemble scheme, we use the prediction results obtained from the other approaches, splitting them into 50% for training and 50% for testing. The configuration of the Bayesian process is similar to that in the sinusoidal regression experiments, while the number of boosting rounds is higher, with 250 steps.

The MAE results for VR Contents 1, 2, and 3 are shown in Figure 7, Figure 8 and Figure 9, respectively. The MAEs of each method are shown for different target tasks. For Content 1 (Figure 7), TFS and TOE have the highest MAE values in most cases. It is not clear which one is better than the other. Meanwhile, MAML and JT show similar MAE results, but one is better than the other for any given target task. The ensemble schemes almost always achieve the lowest MAEs; this is most clearly observed for the yaw direction. For Content 2 (Figure 8), TFS and TOE again have the highest MAE values in most cases. The ensemble schemes show the lowest MAEs in the pitch and roll directions but are worse than MAML in the yaw direction except for target tasks 4 and 7. For Content 3 (Figure 9), which involves the highest level of motion among the different contents, it is clear that the ensemble schemes achieve the best overall performance. The gains of the ensemble schemes differ by the target task since the distributional gaps between the meta-tasks and each target task differ. Meanwhile, inconsistent results are observed for some tasks, in particular for target task 6 for Content 1, target task 2 for Content 2, and target task 6 for Content 3, all for the yaw results, for which TFS or TOE outperforms the ensemble schemes. In general, VR motion tasks show high diversity between tasks as well as large fluctuations within each task. Thus, inconsistent results may appear if a target task has a significantly different structure from the meta-tasks and no meaningful information or even a negative effect is obtained from meta-/pretraining.

In summary, as shown from these experimental results, the ensemble schemes were the best in 75% of the cases, while MAML achieved the best results in 22% of the cases. TOE was the best in two cases, namely, for target task 6 for Contents 1 and 3, but with a very small gain with respect to the ensemble schemes. There were many cases in which JT outperformed MAML; however, in these cases, the ensemble schemes were better than JT. Since in general, the distributional gap between a set of meta-tasks and a target task is not known a priori, using the proposed ensemble schemes, which can adapt well to a wide range of target tasks by combining multiple methods, is a recommended strategy.

### 6.3. Temperature Forecasting

For this regression problem, a temperature dataset from Seoul, South Korea, provided by the Korea Meteorological Administration (KMA) [35] is used to predict the next-day maximum and minimum temperatures. The input data include the per-day maximum and minimum temperatures, the Local Data Assimilation and Prediction System (LDAPS) prediction, the longitude and latitude of the measurement location, evaluation, slope, and solar radiation. The output consists of the next-day maximum and minimum temperatures. The dataset consists of July and August data from 2013 to 2017 captured by 25 automatic weather stations (AWSs) operated by KMA, with a total of 310 data samples for each AWS.

For the temperature forecasting experiment, we select the data from one AWS as the target task and the data from the rest as the meta-tasks. We show the result plots for the cases with AWSs 1, 3, 6, 8, 11, 13, 16, 18, 21, and 23 as the target tasks. We create an NN with four dense layers and one output layer, with a linear activation function for each layer. The number of iterations is 100 for both meta-training in MAML and pretraining in JT. In MAML, the learning rate for the outer loop is set to 0.005, and that for the inner loop is set to 0.001. The number of adaptation iterations is ten. In RF and GB, the prediction results from the different approaches are split into 50% for training and 50% for testing. In the BO process, init_point is set to three steps, and n_iter is set to ten steps.

The prediction results for the next-day minimum and maximum temperatures are shown in Figure 10 and Figure 11, respectively. The prediction error is measured as the MAE. For both maximum and minimum temperature prediction, the ensemble schemes outperform the other methods for all considered target tasks. Of the ensemble schemes, RF performs slightly better than GB, although the performance gap is not significant. The MAE values achieved by the ensemble schemes fluctuate less between different target tasks than those of the other methods, indicating that the ensemble schemes may consistently produce more stable results for diverse target tasks. MAML achieves the next best prediction performance, followed by JT. TFS is the worst, showing significantly higher MAEs than the other methods.

## 7. Conclusions

We investigated how the differences between the data distributions of the meta- and target tasks impact performance in regression problems and revealed that these differences strongly affect the relative performance of different adaptation methods. Based on this observation, we developed ensemble schemes combining multiple adaptation methods that can handle a wide range of distributional gaps between the meta- and target tasks. We evaluated the proposed ensemble schemes on sinusoidal regression, VR motion prediction, and temperature forecasting problems and demonstrated that they achieve the best performance among the considered methods in most cases.

The study reported in this paper has the following limitations: (1) only the basic MAML and transfer learning frameworks are considered, and (2) only regression problems are considered. Therefore, future work should aim to overcome the abovementioned limitations, i.e., future work should consider advanced MAML and transfer learning frameworks as well as other types of problems, such as classification and reinforcement learning.

## Figures and Tables

**Figure 1 sensors-23-00583-f001:**
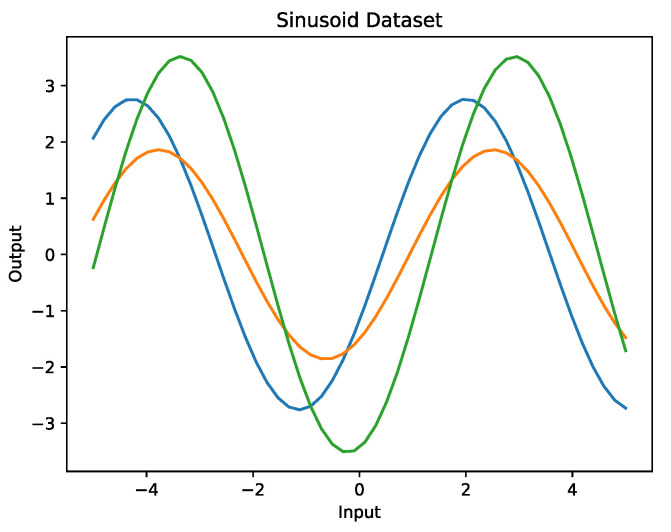
Sinusoidal regression problem.

**Figure 2 sensors-23-00583-f002:**
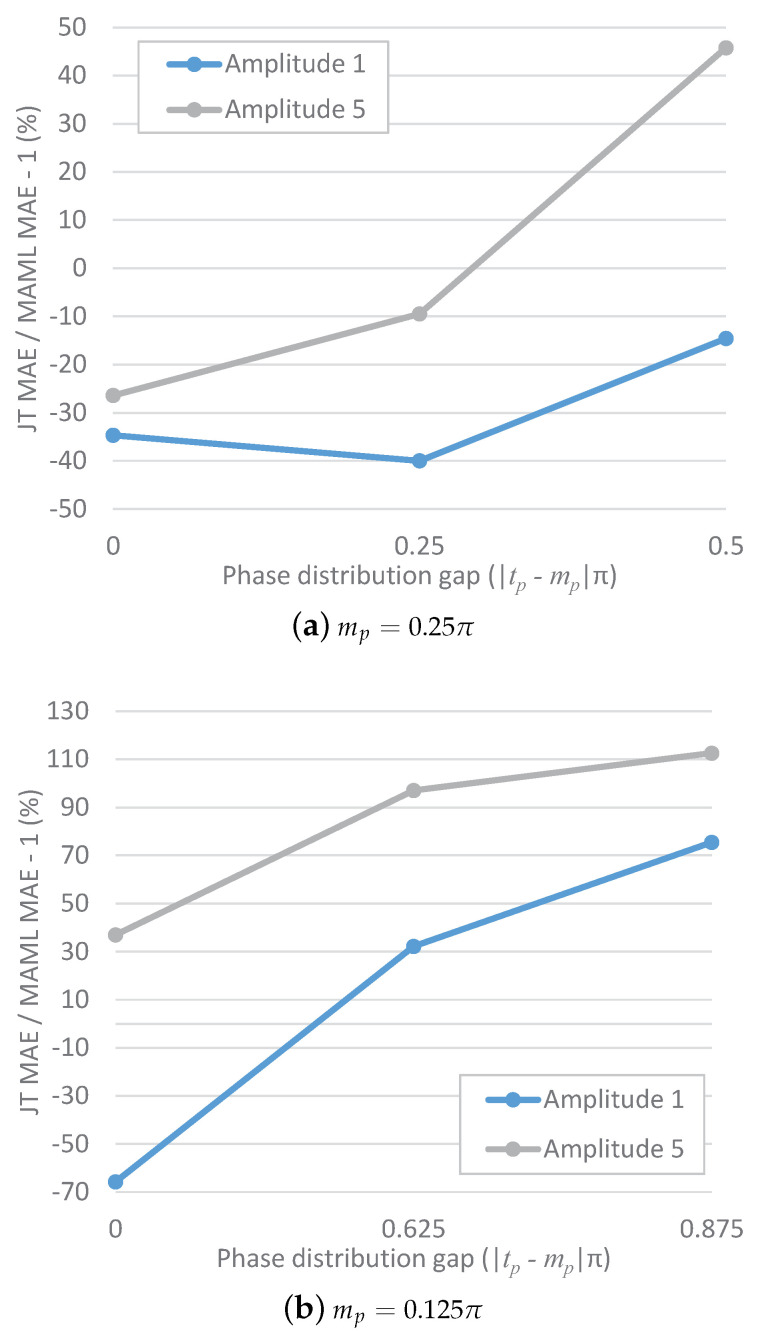
MAE gain of JT over MAML for an increasing phase distribution gap between the meta-tasks and the target task.

**Figure 3 sensors-23-00583-f003:**
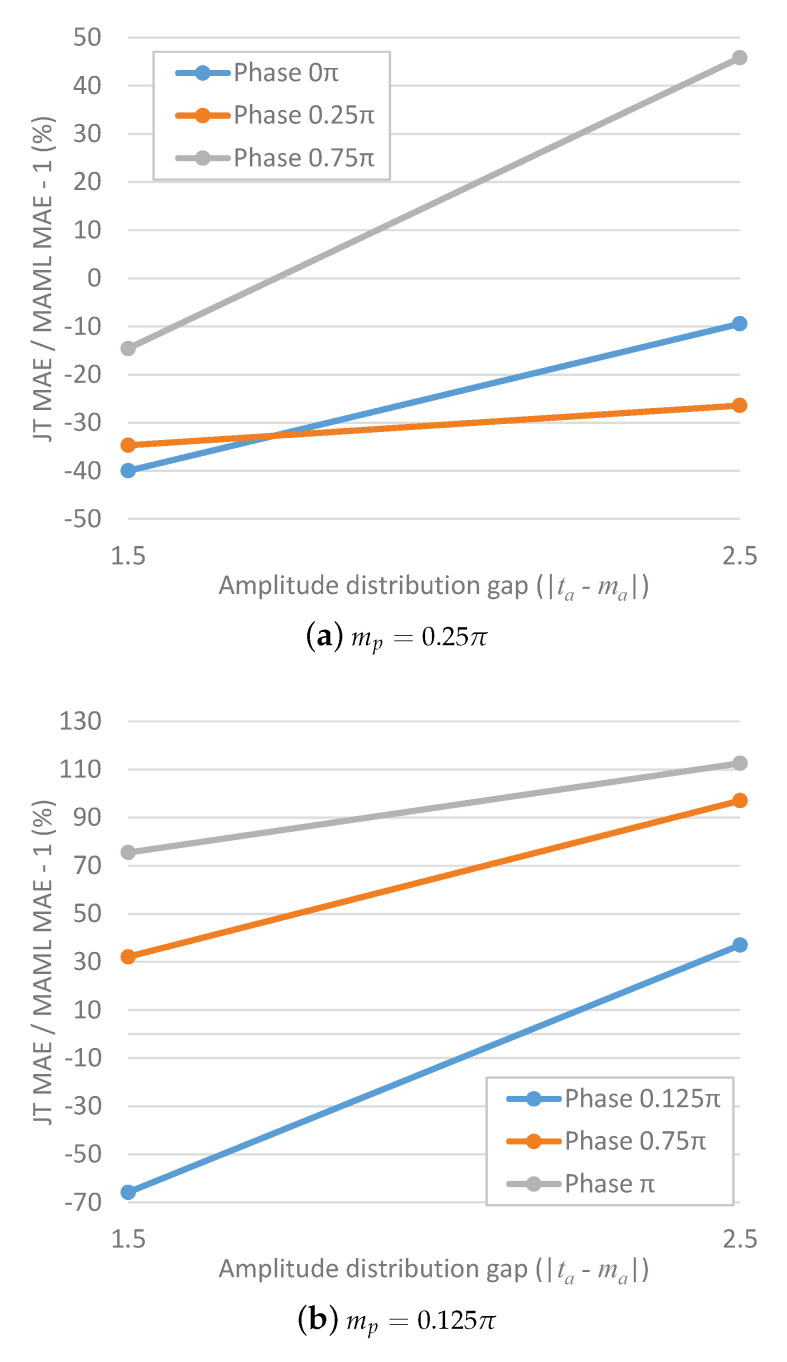
MAE gain of JT over MAML for an increasing amplitude distribution gap between the meta-tasks and the target task.

**Figure 4 sensors-23-00583-f004:**
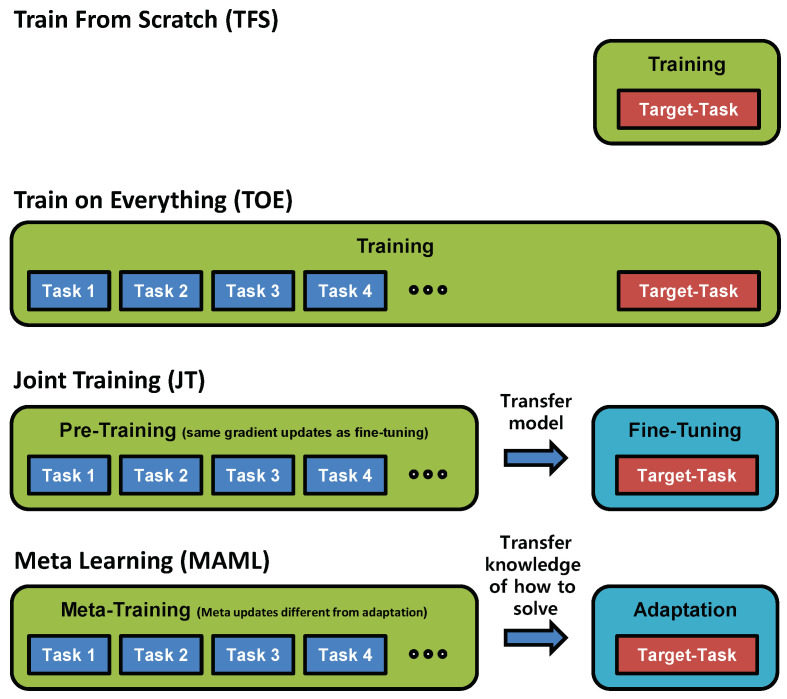
Comparison of training processes and datasets between adaptation methods.

**Figure 5 sensors-23-00583-f005:**
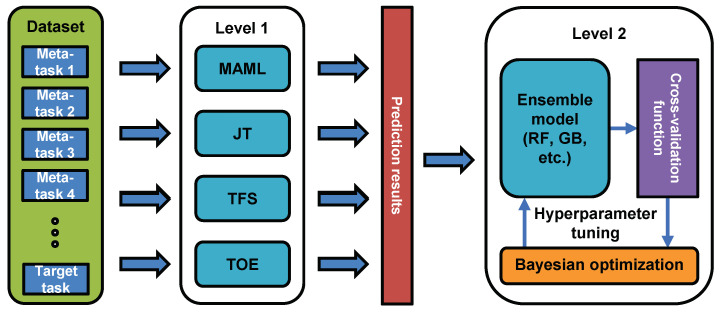
Ensemble scheme architecture.

**Figure 6 sensors-23-00583-f006:**
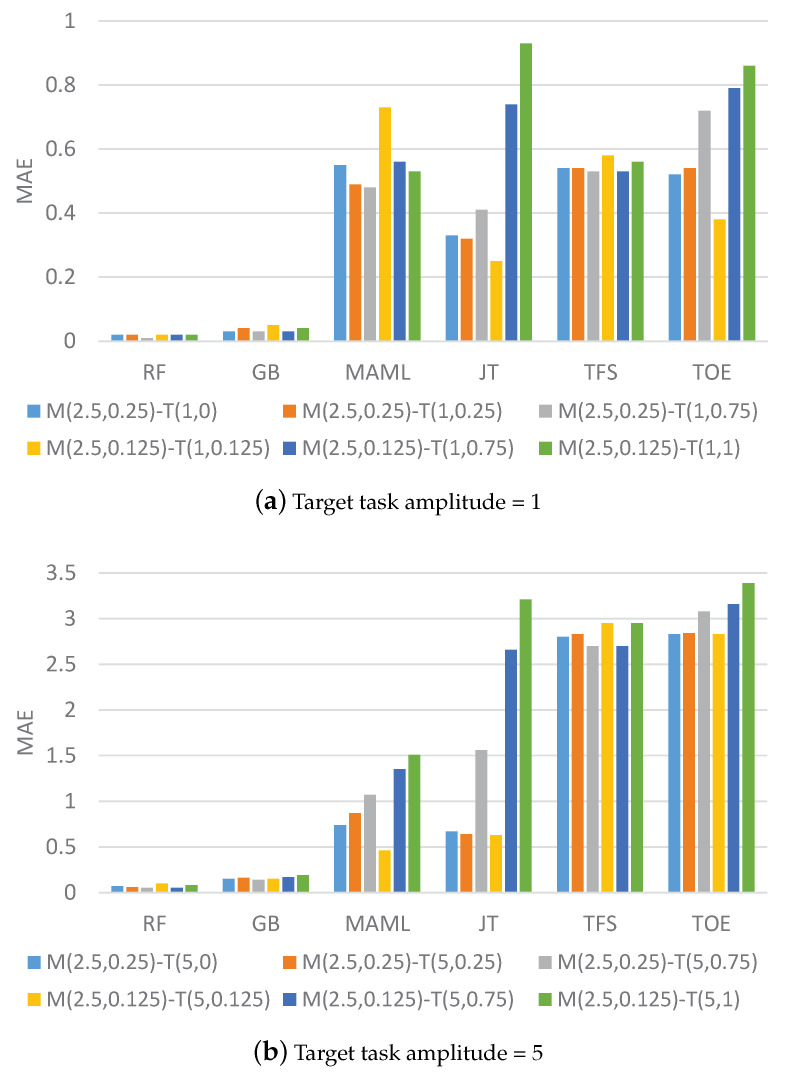
MAEs of sinusoidal regression under different data distribution settings for the meta-tasks and different amplitudes for the target task.

**Figure 7 sensors-23-00583-f007:**
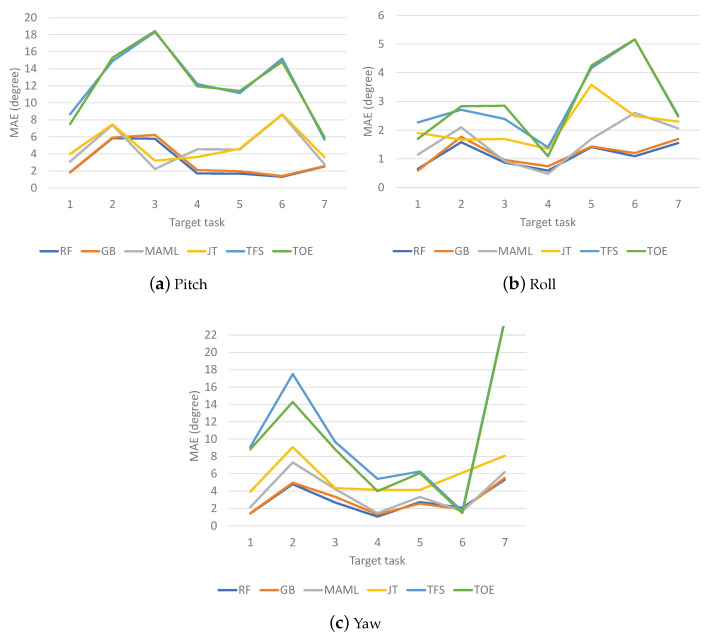
MAEs of VR motion prediction for Content 1.

**Figure 8 sensors-23-00583-f008:**
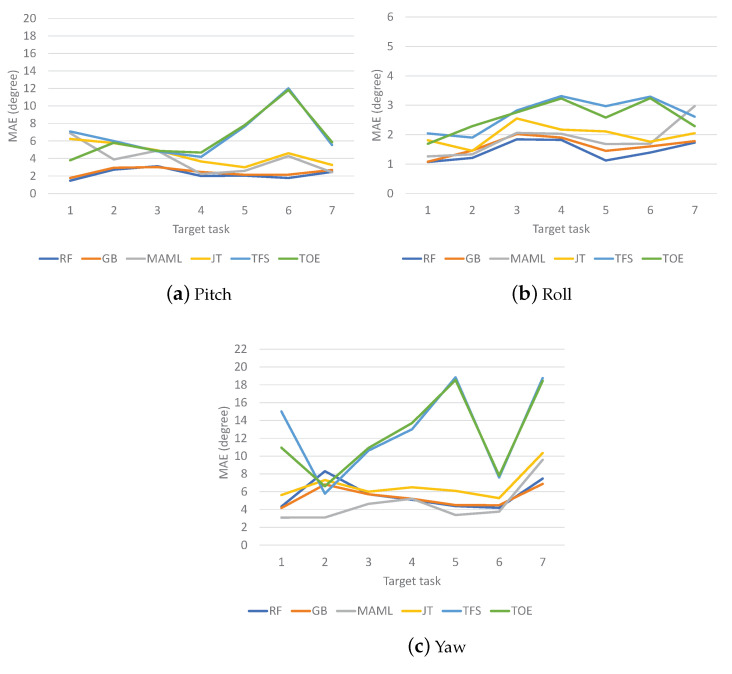
MAEs of VR motion prediction for Content 2.

**Figure 9 sensors-23-00583-f009:**
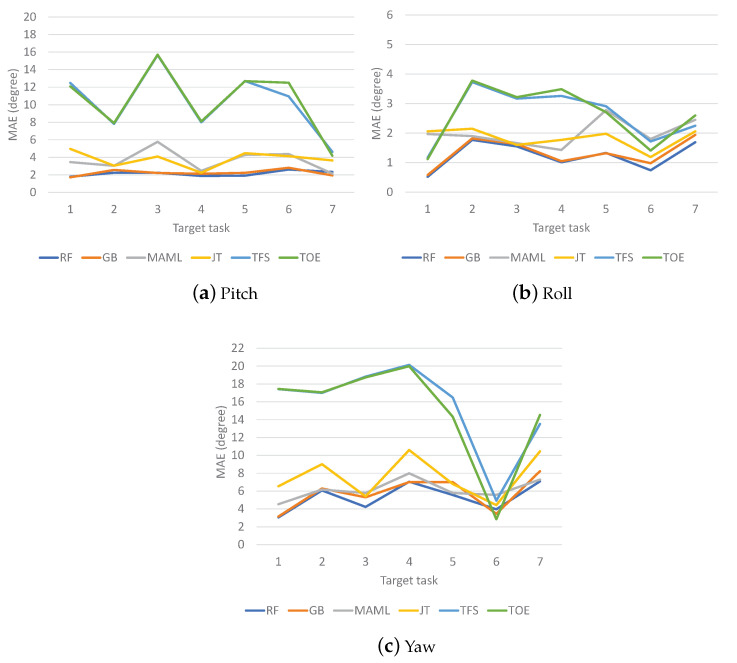
MAEs of VR motion prediction for Content 3.

**Figure 10 sensors-23-00583-f010:**
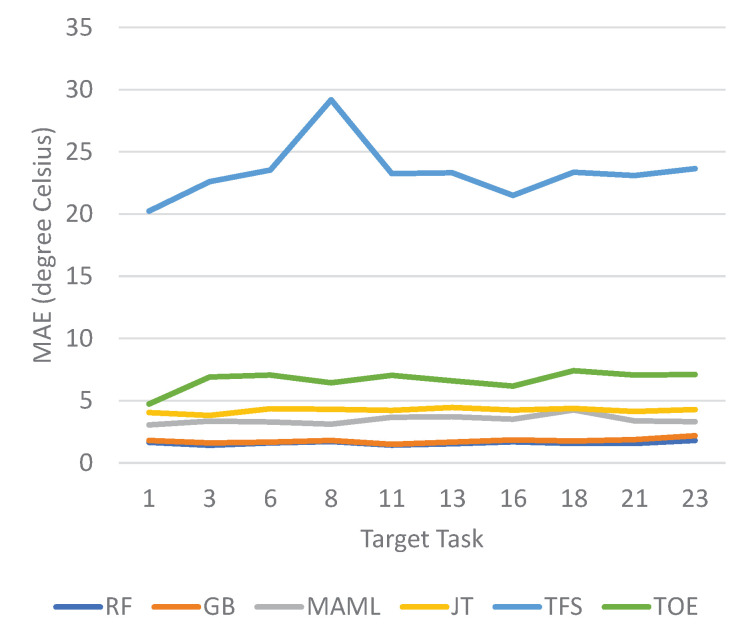
MAEs of temperature forecasting for the next-day minimum temperature.

**Figure 11 sensors-23-00583-f011:**
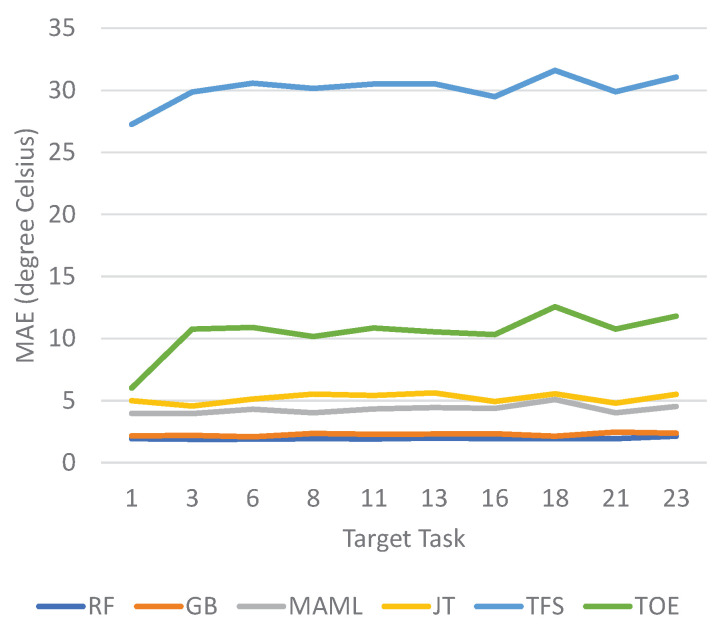
MAEs of temperature forecasting for the next-day maximum temperature.

**Table 1 sensors-23-00583-t001:** Experimental settings for sinusoidal regression.

ine **Setting**	**Meta-Task Amplitude and Phase Distributions**	**Target-Task Amplitude and Phase**
ine M(2.5, 0.25)-T(1, 0)	[1, 5]; [0, 0.5π]	1/5; 0π
M(2.5, 0.25)-T(1, 0.25)	[1, 5]; [0, 0.5π]	1/5; 0.25π
M(2.5, 0.25)-T(1, 0.75)	[1, 5]; [0, 0.5π]	1/5; 0.75π
M(2.5, 0.125)-T(1, 0.125)	[1, 5]; [0, 0.25π]	1/5; 0.125π
M(2.5, 0.125)-T(1, 0.75)	[1, 5]; [0, 0.25π]	1/5; 0.75π
M(2.5, 0.125)-T(1, 1)	[1, 5]; [0, 0.25π]	1/5; π
ine		

## Data Availability

Not applicable.

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
