# Peer review of "Combining Model-Agnostic Meta-Learning and Transfer Learning for Regression"

_sensors, 2023, doi:10.3390/s23020583_

Round 1
Reviewer 1 Report
Please refer to the attachment for detail comment.

Reviewer 2 Report
This paper describes an ensemble scheme to combine multiple adaptation methods and applies the scheme to data regression. Overall, the paper does appear to provide some valuable results, especially in its experiments. However, there are some flaws needed to be addressed and I will list them below. Therefore, my recommendation is a major revision. The issues are listed below:
(1) The second section on related works is very weak. The first issue is that the discussed literatures are too old. Meta-learning, transfer learning, and ensemble learning are all very active research fields but only works before 2018 are presented. The second issue is that transfer learning is not reviewed at all in this section.
(2) In the fourth section and its following sections, using index “i-j” to denote settings is not very informative. Since the main difference in data distribution is in phase but not in amplitude, distinguishing setting by amplitude doesn’t seem to make sense. This also makes interpreting the results in experiments counter-intuitive.
(3) Following the second issue, varying only phase of the sinusoid makes the experiments too limited. A broader range of amplitudes and frequencies should also be tested so that the full spectrum of the problem of sinusoidal function regression is examined.
(4) Following the third issue, the frequency of the sine wave is never revealed in the discussion.
(5) Both Figure 3 and Figure 4 show an upward trend for all curves. This is not discussed in the text.
(6) In the experiments, only one ensemble scheme: XGBoost is tested. What about the other ensemble scheme?
(7) The scales of the 6 bar graphs in Figure 6 are different. This makes comparison unfair. It is better to integrate all these bars together into one graph.
(8) The VR motion prediction experiment shows that XGBoost has its limitations. However, the discussions on these limitations are rather sketchy. The authors should elaborate more on these issues.
Round 2
Reviewer 2 Report
The authors have made all the necessary changes. I've no further comments.